# “Physical Activity Is Not the Answer to Everything, but It Is to a Lot”: Stakeholders’ Perceived Determinants of Implementing Physical Activity Interventions for Older Adults

**DOI:** 10.3390/geriatrics9050113

**Published:** 2024-09-04

**Authors:** Janet M. Boekhout, Rieteke Hut, Jannique G. Z. van Uffelen, Gesa Czwikla, Denise A. Peels

**Affiliations:** 1Department of Health Psychology, Faculty of Psychology, Open University, 6401 DL Heerlen, The Netherlands; rieteke.hut@ou.nl; 2Department of Movement Sciences, KU Leuven, 3000 Leuven, Belgium; jannique.vanuffelen@kuleuven.be; 3Department of Social Epidemiology, Institute of Public Health and Nursing Research, University of Bremen, 28359 Bremen, Germany; gesa.czwikla@uni-bremen.de

**Keywords:** physical activity, older adults, intervention, implementation

## Abstract

Although many physical activity (PA) interventions for older adults have proven effective in controlled research settings, optimal implementation in real life remains challenging. This study identifies determinants perceived by stakeholders when implementing community-based PA interventions for older adults. We interviewed 31 stakeholders guided by the Consolidated Framework for Implementation Research (CFIR). Results showed that stakeholders are very specific about the role they can play in implementation, making collaboration between stakeholders crucial. Barriers and motivators were identified in the CFIR intervention characteristics domain (relative advantage, complexity and costs, evidence quality and strength, and adaptability and trialability), in the outer setting domain (cosmopolitism, patient needs, and external policy and incentives), in the inner setting domain (implementation climate, relative priority, compatibility and organizational incentives and rewards) and in the individual characteristics domain (knowledge and beliefs, and other personal attributes). An overarching theme was the stakeholders’ emphasis on aiming for broad health goals in interventions, as they perceive PA as a means to reach these goals rather than an end in itself. Another overarching theme requiring attention in future implementation efforts is the need to tailor implementation efforts to the specific needs of older adults as the end users of the intervention.

## 1. Introduction

The detrimental effects of insufficient physical activity (PA) on physical, mental, and social health are well-documented [1,2] and come with high costs for society. It is estimated that by the year 2030, insufficient PA will account for 70% of healthcare expenditures [3]. Some groups in society are especially prone to insufficient PA, such as older adults [4]: prevalence data indicate that the large majority of older adults are not meeting the WHO recommendations for sufficient PA and that insufficient PA among this target population has even increased in the last decades [3,5,6]. Numerous PA interventions for older adults have been developed and proven effective in Randomized Clinical Trials (RCTs) [7,8,9]. Thus, because of the increase in the proportion and number of older adults, as well as the health impact of regular PA, there is a large potential for a significant public health impact of PA promotion in this population group. 

The impact of PA interventions on public health is, however, not only determined by their efficacy but also by their implementation in real-life settings. Previous research has demonstrated a substantial gap between scientific knowledge derived from RCTs and public health practice with regard to implementing PA interventions [10,11,12]. In public health practice, stakeholders play a crucial role in implementing PA interventions, as they deliver the intervention from the developer to the end user (i.e., the target population) [13]. As the implementation of (preventive) PA interventions in public health practice is often not primarily concentrated in the health sector, the presence of multiple stakeholders makes implementation a complex and challenging process [14]. It often requires stakeholders at multiple levels, each playing different roles within the implementation process. These stakeholders can come from different sectors (e.g., the healthcare sector, municipal domain sector, sports sector, and other public or even private sectors). Within each sector, different organizations can play a role. e.g., within the municipal domain sector, the municipality itself and the regional health counselor or a welfare organization can play a role. Within organizations, different individuals can have a role (e.g., in healthcare organizations, there are healthcare managers and nurses, and in municipalities, you can have local policy advisors and mayors). No single sector has sole responsibility for all actions needed to increase population levels of PA, nor do the benefits of increasing PA accrue to just one sector [3]. Intersectoral actions and multisectoral partnerships between different stakeholders are urgently needed to successfully implement these interventions [15]. It is, therefore, essential to gain insight into the role that stakeholders from different sectors are able and willing to play in implementing PA interventions, as well as the sector-specific determinants of implementation that stakeholders perceive as relevant. In the current study, we will refer to stakeholders as individuals working within stakeholder organizations.

A well-established scientific framework that is frequently used as a tool for the assessment of implementation determinants is the Consolidated Framework for Implementation Research (CFIR) [16]. CFIR consists of a set of well-described implementation constructs that can influence implementation and, ultimately, whether interventions shown to be effective in RCTs will be used in real-world settings. These constructs are grouped into five domains: (1) Intervention characteristics (e.g., relative advantage and costs), (2) inner setting (organizational characteristics, e.g., capacity and relative priority), (3) outer setting (environmental characteristics, e.g., external policies), (4) characteristics of the individual (person engaged in the implementation initiative, e.g., attitude and knowledge), and (5) implementation process (e.g., planning, execution, and evaluation of the implementation). Cooper et al. (2021) recently systematically reviewed the determinants of implementing community-based PA interventions across all age groups [14]. They identified a relatively even distribution of implementation determinants across the five domains, indicating that, indeed, a wide range of factors might be relevant when implementing PA interventions. More specifically for older adults, a scoping review of Scherpenseel et al. [17] on implementation determinants of fall prevention interventions targeting community-dwelling older adults also identified a broad range of determinants influencing the implementation. Both reviews highlight that successful implementation of PA interventions in the community is challenging for all age groups, including older adults since there is not one single factor that can be identified as a key implementation determinant. Moreover, these reviews point out that much implementation research has focused on interventions with other behavior than PA and on a wide range of end users, while it is likely that stakeholders’ perceptions are dependent on the type of intervention and end user [16], in this case, PA interventions for older adults.

Despite the aging population and the potential impact of PA interventions in this rapidly growing population group, studies identifying implementation determinants, especially regarding PA interventions targeting the PA behavior of community-dwelling older adults, are still scarce. More insight into these determinants in this specific setting is highly relevant as this could provide valuable information for the selection of effective implementation strategies [18], thereby increasing the potential public health impact of proven effective PA interventions. The aim of this study, therefore, is to assess the determinants perceived by a broad group of stakeholders as relevant for the implementation of PA interventions for community-dwelling older adults. These insights are essential in increasing PA in older adults and thus may ultimately contribute to public health.

## 2. Materials and Methods

This study uses a qualitative design: recent reviews have highlighted that research to date on this topic has often been quantitative, whereas qualitative studies could provide much-needed insights into what stakeholders consider to be determinants of implementation [14,17]. In this study, with semi-structured interviews, we adhered to the Consolidated Criteria for Reporting Qualitative Research [19]. 

### 2.1. Participants and Setting

Stakeholders were recruited by referral sampling: in referral sampling, participants are asked to recommend organizations that are particularly relevant, focusing on those with specific expertise or involvement in the area of interest [20]. The interviews were conducted in two provinces of The Netherlands (Limburg and Brabant): two of the researchers/authors involved in this study work in this region, and the other researchers work in different regions of The Netherlands or abroad. The region where the interviews are conducted is largely comparable to the majority of Dutch provinces regarding demographic distributions, with only a slightly higher than average age distribution. Apart from the region, no other exclusion criteria were applied a priori. Emails, including a description of the research project and an informed consent form, were initially sent to seven existing contact persons within the professional network of the main researchers. These seven stakeholders were selected as starting points for the interviews, as these are responsible for broad regions and, as such, they have extensive networks and knowledge of other relevant parties. These primary contacts include five regional health councils and two sports development organizations. After each interview, stakeholders were asked to identify other stakeholders that they considered important for the implementation of PA interventions for older adults in The Netherlands. We did not use a traditional snowball method, where interviewees are asked to suggest another person, but we used referral sampling by explicitly asking what kind of other organizations or people with other kinds of roles are important in the implementation of PA interventions. In this way, we aimed to include all potentially relevant stakeholders within a wide range of positions or qualifications. In total, 22 additional stakeholders’ organizations were identified. These newly identified stakeholders were then also invited by email to participate in an interview, including a description of the research project and an informed consent form. 

All participants provided written informed consent prior to participation. Interviews were recorded and transcribed verbatim, with personal identifiers omitted. Interviews were conducted in Dutch, and relevant quotes were translated into English. The transcripts were not sent to the stakeholders for feedback. 

### 2.2. Data Collection

The (all female) research team comprised two main researchers (JMB and DAP), one junior researcher (RH), and two student assistants (KvA and LS). JMB and DAP had PhDs in Health Psychology and worked as assistant professors in Health Psychology; RH held a master’s degree in Health Psychology and worked as a research assistant. KvA and LS were in the final year of their master’s studies in Health Psychology. JMB, DAP, and RH had prior experience in conducting qualitative research.

The interviews took place between February and June 2022 and lasted approximately 90 min each. Due to COVID-19 restrictions, they were conducted via video calls on the Teams platform. The first ten interviews were conducted by two researchers: first, the two main researchers performed five interviews together to identify potential issues with the interview guide; in the next five interviews, either the junior researcher or a student assistant did an interview together with one of the main researchers, in order for them to get acquainted with the interview guide. After that, each interview was conducted by one of the researchers. 

A semi-structured interview guide (which can be found in Appendix A) was based on a template of CFIR constructs [21]. This template enabled the selection of constructs and questions relevant to this study, which were translated into Dutch by the main researchers. The complete draft of the interview guide was checked by a third person (MK) who was not involved in the current research but had broad experience in organizational change and with performing interviews within the municipal domain. Consequently, some final changes were made in terminology to better fit the perceptions of the stakeholders. The main topics in the interview guide consisted of four of the five CFIR domains: the domain on the implementation process was omitted as this domain focuses on specific implementation processes that have already occurred, which was not the case in our research. For the remaining domains (i.e., intervention characteristics, inner setting, outer setting, and characteristics of the individual), JB and DP adapted the interview guide to fit this research project. For each of these domains, the main questions and several potential questions that could be asked depending on previous answers were formulated. Examples of main questions were “What is your opinion on PA interventions for older adults that are delivered online?” (intervention characteristics); “To what degree does implementing PA interventions contribute to your organizations’ goals? (inner setting), “to what degree do you expect that other organizations support you when you implement interventions?” (outer setting); and “how important do you personally think that stimulating PA in older adults is?” (individual characteristics). A priori to these questions, an introducing question was formulated, in which participants were asked whether they perceived themselves as being able to play a role in implementing PA interventions, and if so, what kind of a role. 

In this study, we did not use a strict definition for older adults, such as defining older adults as 65 years and over [22]: in the interviews, we asked the stakeholders for their opinions about PA interventions for older adults, thus letting them implicitly decide what older adults were. Questions were formulated neutrally, and suggestive questions were avoided. Some questions were omitted from the original CFIR interview guide, as these topics were not relevant to the Dutch situation, and further questions were added. Although the interviews took place during the COVID-19 period, we focused on a normal non-COVID situation.

When discussing the domain of intervention characteristics, we used a proven effective online computer-tailored PA intervention as an example [23,24,25]. This intervention was chosen as a good representation of a scientifically developed and effective intervention capable of large-scale implementation. It also illustrates a type of intervention that may not immediately come to mind for all interviewed stakeholders, who may have previously focused on small-scale local programs; moreover, it was developed at the university of the authors, making them knowledgeable about this intervention.

After 31 stakeholders (within 23 stakeholder organizations) were interviewed, no other relevant parties that also needed to be interviewed were suggested by the already interviewed stakeholders, and data collection was completed. Additionally, based on preliminary findings discussed by RH, JMB, and DAP, it was found that no new information had emerged from the last six interviews, and therefore saturation was deemed to be reached. 

### 2.3. Data Analyses

A preliminary codebook was developed a priori by KvA, RH, JMB and DAP (see Appendix A), based on the existing codebook available on the CFIR website [21]: this codebook was adapted to the specific Dutch situation. 

Atlas.ti was used to manage the data [26]. Data were analyzed using deductive thematic analysis [27] consisting of the following steps: (1) familiarization; (2) data coding; (3) generating initial themes; (4) reviewing and developing themes; (5) refining, defining, and naming themes; (6) writing the report. KvA and LS coded the interviews from one province each. All coding was reviewed by RH. When RH differed in opinion on specific coding, this was discussed with KvA and LS. When no consensus was reached, JMB or DAP were included in the discussion. Themes were generated based on the codebook by KvA, LS, RH, and JMB. It was constantly checked whether themes other than those from the pre-defined codebook had arisen from the data. JMB wrote the manuscript; the other authors reviewed and revised the manuscript. To protect the privacy of the stakeholders that participated, we did not specify the type of organizations that had delivered each quote. Instead, we only assigned a number to each participant. Given the roles of organizations in our specific region, revealing this information could make it possible to deduce who made the statement.

## 3. Results

We interviewed 31 stakeholders within 23 organizations, 15 interviews with one stakeholder, and eight interviews with two stakeholders (from the same organization). The stakeholders came from 10 different types of organizations, i.e., regional health council (5), municipalities (3), sports development organizations (8), lifestyle coach (1), senior citizens organization (1), wellbeing organization (1), healthcare organization (1), physiotherapist (1), public housing (1), general practitioner’s network (1). Six additional organizations were invited to participate but declined for various reasons, such as lack of time or interest. Eleven organizations were mentioned by other stakeholders but were not invited because they were in another region where our interviews were held (8) or because they did not focus on independently living older adults (3). The latter was not established as an exclusion criterion a priori, as we aimed to focus on PA interventions for community-dwelling older adults. However, some stakeholders believed our research also targeted institutionalized older adults, which was not our intention. Therefore, it was decided to exclude these types of organizations when they were proposed. 

### 3.1. Stakeholders’ Roles in Implementation

Each interview started with the question of whether participants saw a role for themselves in the implementation of PA interventions and, if so, which role. When stakeholders asked for suggestions on what roles we meant, we suggested developing, financing, organizing, coordinating, facilitating, or informing. All stakeholders expressed that they perceived a specific role for themselves in the implementation of PA interventions. Stakeholders indicated that they must fulfill their individual roles in alignment with the goals and potential of their organizations. They expressed that even if they would like to perform other roles within the wide range of roles and responsibilities that are needed when implementing interventions, they are constrained by their job descriptions and their organization’s objectives. Consequently, they emphasized the importance of a good network to fulfill all tasks that are needed to implement an intervention. “*My network is my gold*”. (#4) 

Only municipalities expressed that they would be able to play a financial role, but most organizations see a role that is either organizing, coordinating, facilitating, or informing. The majority of stakeholders indicated that they did not consider the development of interventions to be an appropriate role. Their attitude was that there are numerous PA interventions already in place, and therefore, there is no need to develop or even adopt new ones. *“There are already so many proven effective interventions, so for me, there is no need to develop new ones”.* (#23) 

There was a striking discrepancy between the perspectives of general practitioners and other types of stakeholders, most of which believed that general practitioners should play a major role in implementing PA interventions. In contrast, the general practitioners themselves did not share the views of the other stakeholders. The general practitioners acknowledged their ability to encourage individuals to be more physically active. However, they perceived a lack of opportunity for themselves in large-scale implementation projects due to time constraints and resource limitations. *“A general practitioner can be a good source for the target population to confront them with their behavior (……) but they are also like ‘hey, we’re doctors, not social workers’. If a general practitioner knows who should be directed to where then they can do that, but they also get a lot of requests for which they have no solutions. They find it important, but the gap between the question and the solution is sometimes very wide”.* (#26)

### 3.2. Domains

Below, we present the findings per CFIR domain. We organized results per domain in line with the determinants, such as described on the CFIR website, for the intervention characteristics [28], the inner setting [29], the outer setting [30], and the individual’s characteristics [31].

#### 3.2.1. Domain 1: Intervention Characteristics

Regarding the determinant *Relative Advantage,* it was found that although stakeholders were positive about PA interventions, they preferred interventions that did not have PA as the sole goal. They preferred interventions that either target broader health-related goals, such as healthy lifestyle and stress management, or interventions that use PA as a means to achieve other goals, such as reducing loneliness. *“PA is not the answer to everything, but it is to a lot. Bringing people together, creating cohesion in a neighborhood, loneliness, risk of falling, self-reliance, vitality … An intervention having a proven relation with positive health is a prerequisite”.* (#7)

Several findings related to the determinant of *Evidence Quality and Strength.* Proven efficacy did not seem to be an essential characteristic of interventions for many stakeholders, nor was it relevant to who had developed the intervention. Although proven effectiveness can make it easier to obtain funding for an intervention, stakeholders mentioned that other ways of demonstrating that an intervention is of good quality are also acceptable. “*Instead of having to read an entire Methods and Discussion of a research project, just clearly communicating ‘this has happened, and these are the outcomes, and therefore we think that this intervention works well’ that would help to get convinced”.* (#14)

A dominant view was that proven effective interventions often have the disadvantage of not being adaptable to the local situation or to the end users, whereas this adaptability was considered very important among stakeholders. *“The thing with proven effective interventions is ’It has to be done like this because this is what we have demonstrated, and therefore you can’t deviate from it because if you do, then we don’t know if the intervention will still be effective’, that for me is a drawback of proven effective interventions. But at the same time, I believe that if we do more of the things of which we know that they work, then we’ll get closer to the solution… so that’s a tricky balance”.* (#1)

All stakeholders mentioned that resources are always scarce, so intervention costs should be kept as low as possible or in proportion to the number of end users reached. In contrast, some stakeholders pointed out that the quantity of people that an intervention can reach holds less significance compared to the demographics of the people reached. Specifically, they emphasized the significance of interventions that are able to include inactive or hard-to-reach populations. *“If I reach 300 participants, of whom 295 exercise three times a week, is that a success? Or do I rather reach 10 that are not active and become structurally physically active? That, for me, is a bigger success than reaching 295 people”.* (#20)

Some stakeholders regretted that interventions are often selected based on short-term gains rather than on long-term gains, such as a reduction in public healthcare costs, which stakeholders consider more important. *“There are numerous studies that demonstrate that every euro invested in prevention in a good way will more than pay for itself. But that’s a short-term, long-term problem. The long-term benefits, you don’t really see, just because people don’t consult you anymore”.* (#7) As these stakeholders expect that long-term gains are difficult to assess, they feared that interventions may not be continued after the first implementation. Therefore, they preferred qualitative assessments of results over quantitative ones, as the first may give an indication of the results before quantitative measures. *“Effects for me aren’t always really measurable. You always hope that the intervention triggers some sort of movement. That you see things happen or that an organization says, ‘Yes, we find this important, and we are going to implement this ourselves”.* (#9)

The determinant of *Complexity and Costs* was reflected in the fact that several stakeholders expressed that interventions should not be too complex or labor-intensive to implement: *“If you see the implementation manual, you already get a little bit tired, you have to do this, and you have to do that… it just seems like a hell of a job, you have to really delve into it, and you need so many volunteers and a project manager….”.* (#6)

Interventions that can be tested in small pilots first and then implemented on a larger scale if they are successful were preferred. “*First, you try it out on a small scale in a pilot; does it work, is this what we want? And when the end users are super enthusiastic, then you start implementing on a bigger scale. That’s how we deal with innovations”.* (#26)

The determinants of *Adaptability and Trialability* were also relevant to many stakeholders. Being able to offer a range of different interventions was considered more important than the relative advantage of one intervention over another, as stakeholders want to reach as many as possible, especially those end users who are hard to reach. *“That’s why it’s so important to have different flavors; some you reach with this type of intervention, and another you reach with that type of intervention, and some not at all”.* (#15)

#### 3.2.2. Domain 2: Outer Setting

Regarding the CFIR determinant *External Policies and Incentives*, many stakeholders noticed that PA in itself is not a priority in external policy. More specifically, PA was considered to be not a goal in itself but a way to reach other goals, such as positive health. Stakeholders indicated that the current shift in Dutch national policies from a focus on older adults to younger generations affects their potential to have a role in the implementation of PA interventions for older adults in several ways. Some mentioned that they are still trying to implement PA interventions for older adults by including them under other goals. *“PA can have priority, but in a different way, in terms of people having to live independently for as long as possible because of shortages in retirement home, so the longer people can stay active, the better not only for the older adult, but also for society”.* (#18)

In addition to national policies, more individual preferences in other organizations can also play a relevant role within local policies. *“An issue we have to deal with are local politics: a new city counselor can have other goals, almost personal preferences. That can conflict with what we have built in our own organization in the years before. And at the end of the day, the one that pays, decides… that how the world rolls unfortunately”.* (#1)

Apart from these external policies, the end users (i.e., the older adults) were the most important factor in the perceptions of stakeholders, which is coherent with the determinant that CFIR describes as *Patient Needs*. Stakeholders believed that it is very important to assess the needs of the end-users before deciding what intervention to implement. Some argued that an intervention should not be spread too broadly but that needs should be assessed area by area or neighborhood by neighborhood, as needs can vary widely. “*What PA or sport interventions are already available in this area, that’s what we chart. Then we perform among the older individuals a needs assessment. And that shows real divergence; per municipality, per neighborhood, the differences are just huge”.* (#15)

Most stakeholders expressed that, in general, the end users are often not aware of the importance of PA, making it difficult to reach them for PA interventions, which is even more pronounced for certain populations. “*The people that are physically inactive don’t know, it’s the hardest group to get going. I don’t know the solution, but approaching them individually and pointing out why PA is important… but let’s not approach thousands of people in a neighborhood because then you only reach those you don’t want to reach”.* (#20)

In terms of the needs of the end user, many stakeholders also believed that the end-user has other priorities than being physically active, such as the need to socialize more with others or other issues in their lives, such as poverty, that may make them uninterested in PA interventions. “*If an intervention delivers on multiple factors, then people may be inclined to participate, so not ‘I have to become active because there is a PA intervention’. No, I’ll go to an intervention because it teaches me what a healthy diet is and how I can deal with loneliness. So, yeah, I think that for the end users, it’s important if they can get more out of an intervention than just PA”.* (#20) 

Apart from knowing what the needs of the end users are and addressing those properly, another feature of the outer setting that the stakeholder consistently mentioned as important is having a robust network and having good alignment within the network, which can be grouped under the determinant of *Cosmopolitism*. “*Do we know from each other what we are doing, what we have? I think there are a lot of interventions that are similar, and that’s not a bad thing, but if we know what we have and what we do, that makes it easier. Working as a chain is important; working together is essential to make a difference”.* (#15) 

In some cases, organizational commitment to actively implement (PA) interventions is lacking due to stakeholders believing that counterparts in other organizations should take this role or should assume greater responsibility. *“We would definitely try to stimulate interventions, but we are not all going to do this ourselves”.* (#8) 

Some stakeholders also admitted that there may be individuals who have an important role in the implementation of PA interventions but who are not known to them or are not included in regular networks. “*I organize a walking intervention in a neighborhood, but when the social worker is on vacation, then suddenly, I have less participants…so that shows how important it is that these people are included in implementing interventions too”.* (#19) 

Financial restrictions were consistently considered to be a limitation. *“Well, that’s a little crooked. Because in the health insurer’s expressions, all of a sudden, it’s all about prevention and combined lifestyle intervention and vitality. But actually, yes, we’ve been wanting that for years. And yes, it’s not being honored”.* (#24)

#### 3.2.3. Domain 3: Inner Setting

Regarding the determinant of *Implementation Climate*, stakeholders all expressed that internal policies are mostly driven by external policies. Currently, many stakeholders have shifted their focus from older adults to youth, or they have to allocate resources to other priorities instead of PA, such as reducing loneliness or improving the local physical environment. This shift in focus in the inner setting is driven by shifts in policy in the outer setting, in this case, Dutch governmental policies. Some policies appear somewhat compartmentalized, and stakeholders regret that there is no connection made between PA, social goals, and health. “*Those nation-wide programs are often directly aimed at municipalities. Municipalities are then facilitated to run these programs; they get funding for that, so that’s what they do… Participation, social connectedness, those are themes that get more attention. Health has zero priority in municipalities, don’t get any illusions on that, and nor does PA.”* (#2) 

Several stakeholders also expressed that the decision to implement may have other motivations than actual internal policies. *“Well, these days it’s also important if a city counselor can score, to put it bluntly… For us, we want to make sure that an intervention works, that it is effective, but that’s not what is important for local councils, for them other things matter”.* (#4) 

Some findings related to the determinants of *Relative Priority* and *Organizational Incentives*. In some organizations, employees, especially those with decision-making power, change every few years, making internal policies unstable and consequently changing the possibilities of implementing interventions. *“Every four years, we get a whole new board. And that new board then starts writing all new policies all over again and sets new goals”.* (#4) 

In order to demonstrate internally that interventions are important and that these interventions are achieving their internal goals, most stakeholders stressed the significance of monitoring the progress towards these goals to ensure that they are being met. Stakeholders thereby seemed to vary in their priorities regarding the metrics needed to assess these goals. For instance, local municipalities may prioritize reaching a large number of end users with interventions, emphasizing quantitative impact. Conversely, organizations like sports development organizations may prioritize qualitative impact by rather delving into the narratives behind the numbers. *“You have to let the data speak, so you have to look at what is going on and what it means for the citizens of our region, and then maybe we will see that we need to shift our focus more on healthy aging.”* (#9) 

The determinant of *Compatibility* was also present from the stakeholders’ perspectives. The limited availability of resources within an organization, both financial and otherwise (such as personnel), can have an impact on what stakeholders can do in implementing interventions. *“When there’s an intervention where I’m asked to contribute, I always have to check in my organization what time it’s going to take and where I can get it from or where I can allocate less time. That is a decision I have to make. It’s not that if there’s an intervention, I can just participate because I always have to check our resources.”* (#20) 

The size of an organization also seemed to be related to the role one can play during intervention implementation. As most stakeholders have relatively small organizational settings, stakeholders find it easy to have contact with others in their organization about interventions or proposals. “*We don’t have a very large department, so we have our people who are involved in the social domain. And they include, for example, the policy officer who is involved with our citizens, our public relations officer. And there you can ask ‘how do you think about it and where is profit in it.”* (#8) 

Stakeholders in bigger organizations mentioned that one department sometimes is not aware of what the other department is doing. All stakeholders indicated that no individual within their organization has the authority to decide whether they can assume a role in implementing interventions. For this decision, authorization from multiple levels of the organization must be obtained.

#### 3.2.4. Domain 4: Individual Characteristics 

*Knowledge and beliefs* were determinants that seemed to be reflected in the stakeholders’ perceptions. During the interviews, when we asked stakeholders about the role they could play in implementing interventions, it became apparent that many were unfamiliar with the term ‘intervention’ and often requested an explanation or definition. Initially, most stakeholders associated interventions with small-scale, locally developed programs conducted in group settings. However, after providing a definition (“proven effective scientifically developed programs aiming at changing individual PA behavior) and an example (as mentioned in the Methods), stakeholders understood what was meant. They generally expressed a positive attitude towards the health benefits of promoting PA and the target population of older adults. They are of the opinion that these end users often had too little attention. *“In my personal opinion, I find it really a pity that important target populations are forgotten. We are not going to get a healthy region if we only focus on youth, especially when older adults are such a large group, and then you just ignore them”.* (#2) 

Within the determinant *Other Personal Attributes*, some other findings could be grouped. Some stakeholders expressed that interventions should not be used to force people into engaging in PA behavior. *“Some people, well, you should just let them be. That may seem strange, coming from someone who holds positive health very high. But if an older person has always lived his life in a certain way and is happy, then I would say, ‘Even if he smokes his cigarette and drinks his pint of beer every day, just let them live on as they like’’.* (#15)

Stakeholders had varied opinions on the ongoing digitization of society and its reflection on PA interventions. Some feared that older adults might not be interested in or able to use interventions with digital components, but others feel that older adults nowadays do not have these issues anymore or that such an intervention could even stimulate them to do more things digitally. *“It’s true that many older adults are not digitally literate. But I see also a lot of older adults who are, or who maybe can ask for help at a library or a grandchild. I’m not afraid of that; after all, it’s how our whole society looks like now”.* (#6) 

Stakeholders believed that if they want to promote PA, they may need to take an approach other than focusing on PA to engage end users in PA interventions. *“What we have in our minds is ‘people should be more active’, but we translate that to them as ‘would you like to join us for a cup of coffee, come join us at…’, so we use our secondary goal to keep the threshold as low as possible”.* (#11).

The findings are summarized in Table 1.

## 4. Discussion

In this study, we utilized the CFIR to gain deeper insight into the determinants perceived by stakeholders when implementing PA interventions for older adults. We found that stakeholders generally identified the same determinants that could either hinder or facilitate implementation or sometimes both, depending on the setting. The results offered a wide range of information, revealing several overarching issues spanning multiple domains. 

### 4.1. Stakeholder Roles in Implementing PA Interventions

A relevant finding of the current study is that although the implementation of effective PA interventions in practice is still very limited, all interviewed stakeholders were willing to play a certain role in the implementation of PA interventions targeting older adults. This is also reflected in the high response rate of stakeholders who were asked to participate in the current study: 26 stakeholder organizations (in total, including 31 individual stakeholder participants) were willing to participate in our research. While each stakeholder mentioned being willing to play a specific role (e.g., financing, facilitation, or organizing), none of the organizations was able to take care of a complete intervention implementation process of developing, financing, organizing, coordinating, facilitating, and informing the end-user, due to restraints within their organizations or job description. Therefore, networking and collaboration with other organizations within and across different sectors were considered particularly crucial for successful implementation. This is in line with the findings of a previous scoping review on the implementation of fall prevention interventions in older adults, which also states that working collaboratively with the right stakeholders, within and outside an organization, is one of the essential implementation determinants. Moreover, the unclarity of roles and responsibilities among involved stakeholders was found to be an important factor in this review [17]. In addition, a recent and extensive pooled analysis study underlines that multi-sectoral efforts are needed to stimulate PA, especially in older adults [32]. The role stakeholders are willing to play in intervention implementation can be related to the different domains of the CFIR, as described below.

### 4.2. Domains

#### 4.2.1. Domain 1: Intervention Characteristics 

In this domain, several determinants were found to play a major role in the implementation of PA interventions for older adults. In terms of the CFIR [28], these can be grouped under the determinants of *relative advantage*, *adaptability and trialability, evidence quality and strength*, *complexity, and costs*. 

For a PA intervention for older adults to have a *relative advantage* over other interventions, it seems that the intervention should not have PA as the sole or primary goal. PA should rather be used as a means to achieve other goals, with a comprehensive understanding of health and the number of co-benefits of PA in mind. In this regard, focusing on promoting social interactions, living independently, and reducing loneliness seems to be particularly promising. First, our results indicate that these goals seem to better address the needs of older adults, increasing their motivation to engage. This matches with previous research, which also shows that interventions that have broader health goals in mind—such as mental health improvement, social interaction, and chronic disease management—are often better appreciated by participants [22,33,34,35]. One issue to consider is that a recent study showed that when communicating about interventions, end users dislike messages that focus on health problems but prefer messages that emphasize the enjoyment that PA can bring or the mental, social, and cognitive benefits of health [22]. Second, our findings show that increasing PA in itself is not a priority in external policy and consequently also often not in internal policies, but the social, economic, and environmental co-benefits of PA promotion are. Although researchers are well aware of the comprehensive health benefits of PA, it is important to recognize that stakeholders may not view PA as a sufficient standalone goal. They understand that PA promotes health but may narrowly associate it with just increasing activity levels. Therefore, when designing and implementing a PA intervention, it is crucial to communicate clearly that PA is a means to achieve a broader spectrum of health benefits, including physical, mental, and social well-being. This approach ensures that the intervention is not merely perceived as promoting more PA but is valued as a holistic strategy for enhancing overall health. To increase the implementation of PA interventions for older adults, intervention developers are therefore challenged to make clear what other goals can be reached with their intervention, being clear of the co-benefits of PA promotion and having both the needs of older adults and the political priorities in mind.

In line with the CFIR determinant *Adaptability*, interventions should be adaptable to the local situation to promote successful implementation. This is in line with the findings of previous reviews indicating that adaptability to the specific context of an intervention is an important facilitator for interventions [36]. In accordance with this, we found that stakeholders perceive the possibility of trialing an intervention as beneficial. When deemed a success, interventions can then be scaled up or adapted to better meet the local needs. Seeking feedback from the end users on their perceptions regarding intervention components and materials can provide valuable insights into potential barriers to intervention engagement and can increase its acceptability and feasibility, which can ensure that the intervention is as likely to be successful in changing behavior as possible [37,38]. This desire for *Adaptability* and *Trialability*, however, may conflict with the determinant of *Evidence quality and strength*. Stakeholders fear that proven effectiveness does not align with adaptability, or they think that these interventions are more complex. Although proven effectiveness is considered important by stakeholders, intervention owners should make it clear which parts of the intervention can be adapted without compromising proven effectiveness [39,40]. Co-creation methods (including universities, stakeholders, and end users) and a deliverable that focuses on adaptation for different settings, therefore, seem to be advisable strategies that have also been suggested in previous research [41,42,43]. 

Regarding the CFIR determinant *Evidence quality and* strength, our study highlights a discrepancy between how the efficacy of an intervention is determined in randomized controlled trials (RCTs) and how stakeholders define the success of an intervention. While intervention developers typically prioritize the effectiveness of significantly increasing PA levels, stakeholders may have different priorities. Although they appreciate the quality assurance that proven effectiveness offers, stakeholders might be more interested in other outcomes. For example, some stakeholders are primarily concerned with reaching a large number of end-users, whereas others find it successful when hard-to-reach populations are reached, such as populations of lower socio-economic positions. Hard-to-reach populations in this regard are especially those with a comparatively high risk of not being sufficiently physically active, such as populations of lower socioeconomic position (SEP) (e.g., low education and income or ethnic minority populations) [44,45,46]. Results from a systematic review and meta-analysis indicate that health behaviors, including PA, play a major role in explaining health inequalities [47]. It has been found that particularly interventions aiming at changing individual behavior may unintentionally increase health inequalities by benefiting high-socio economic positions (SEP) individuals disproportionally more than low-SEP individuals [48,49]. According to the World Health Organization, individually focused intervention approaches to promote PA should be implemented according to the principle of proportionate universalism, i.e., aiming at all groups of older adults but at a scale and intensity that is proportionate to the degree of need [3]. It is obvious that aiming for a large number of participants or aiming for those hard-to-reach will affect implementation choices and outcomes, and therefore, it is important to assess what stakeholders find important in what they want to achieve [50]. It is important to keep in mind that only 40% of those aged 65+ years meet PA guidelines [51]: from a population health perspective, the largest impact can be made by getting large population groups who do not meet guidelines to engage in more PA. In this case, the challenge is to reach those large groups with low PA levels for maximal impact, considering financial and logistical implementation limitations. One issue to consider is that previous research has shown that many older adults believe that they are sufficiently active while actually overestimating their level of PA [22,52], making it important that in intervention recruitment messages, stakeholders should make clear what sufficient PA is. 

The difference in relevance of effectiveness metrics was also identified in one of our previous implementation trials, which showed that although the intervention was effective in increasing levels of PA, implementation was not continued by stakeholders due to the lower-than-expected numbers of participants [50]. This study thereby highlights the importance of performing qualitative research while identifying implementation determinants in order to clarify which outcome measure is perceived as most relevant. 

Stakeholders acknowledge that it is challenging to assess the long-term health effects and potential cost savings of an intervention. The literature from other countries has also demonstrated that these effects are highly context-specific (for example, regarding policies and culture), indicating that a one-size-fits-all approach is inadequate [53]. It is therefore recommended that, when implementing, stakeholders are consulted to ascertain which target groups and outcomes are of greatest importance to them and are feasible to address given budget and personnel in order to be able to chart this information for them in the implementation process. 

The perceived *Complexity and Cost* of implementing an intervention are regarded as barriers to uptake. A good manual may thus be essential; however, a manual that is too extensive seems a deterrent for implementation. It is striking that interventions that are proven to be effective for increasing PA are often perceived as complex to implement. Stakeholders prefer interventions that are easy to implement, have a practical manual, and do not require a lot of resources. It might, therefore, be advisable to identify effective interventions but to pay attention to easy guidelines on practical implementation. Our findings are in line with the previous literature from the UK and USA that also emphasizes the critical role of well-developed, easy-to-understand manuals in ensuring the effective implementation, fidelity, and scalability of PA interventions [54,55].

#### 4.2.2. Domain 2: Outer Setting

According to CFIR [16], stakeholders that are supported and encouraged by other external stakeholders are more motivated to implement interventions and to sustain them for longer. This was also reflected in our findings where, in the terms of CFIR for this domain [30], *Cosmopolitism, Patient needs*, and *External policy and incentives* were seen as the most relevant determinants. 

Regarding *Cosmopolitanism*, the stakeholders’ perceptions make clear that a good network and good alignment within that network are important. This result is in line with a recent review [14], which also showed that poor relationships between organizations and communities and lack of coordination and communication between organizations can hinder the implementation of interventions in a community setting across different age groups, including older adults. As no stakeholder seems to be able or willing to play all roles needed in an implementation process, collaboration between stakeholders and maintaining a strong network seems crucial. This means that when implementing an intervention, including all relevant stakeholders in such a network is key: the literature provides several strategies to facilitate such collaboration [16], for example, by promoting network weaving (e.g., to promote information sharing, collaborative problem-solving) or by creating a learning collaborative. As settings and relevant stakeholders may change over time, it is important to regularly check whether all stakeholders are still included. This interactive process was endorsed previously [50].

In line with the CFIR determinant of *Patient needs*, insight into the needs of the end-user was found to be the most important implementation determinant of the outer setting. These needs differ between different population groups of older adults. For example, there may be differences regarding motivational factors and barriers to perceptions about and attitudes toward PA, preferred PA domains, and cultural expectations [56,57,58]. The review by Cooper et al. (2021) also showed that community involvement in support of the intervention was an important facilitator for implementation [14]. However, these *Patient needs* are very context-specific [59], and therefore, it seems essential to assess the needs of the end-users before implementing an intervention [60,61,62], e.g., using focus groups with representatives of the end users or other participatory approaches. A potential barrier mentioned by stakeholders is that many end users are unaware that they are not physically active enough and, therefore, do not seem to notice information about PA interventions, or they have other priorities [63]. Attracting end users by appealing to other priorities that the end user has, such as the opportunity to socially interact with peers, seems an interesting option [64,65]. The existing literature has demonstrated the importance of integrating the perspectives of stakeholders and end users when adapting interventions [66,67]. 

Stakeholders perceive a shift of national policies from older adults to youth and from PA to more societal issues such as social connectedness. While the latter shift appears to be a global trend [68], the former shift (from older adults to youth) seems to be particularly evident in the Netherlands and possibly also in some other countries. The review by Cooper et al. (2021) [14] also identified the instability or lack of *external policy and incentives* supporting interventions for older adults as an important barrier to implementing interventions. 

#### 4.2.3. Domain 3: Inner Setting 

The CFIR states that the extent to which an intervention is supported within an organization impacts its chances of implementation. At this inner setting level, our findings show that in terms of CFIR [29], the determinants of *Implementation climate*, *Relative priority*, *Compatibility,* and *Organizational incentives and rewards* were the main factors that influenced implementation. 

A matter to consider in the *Implementation climate* is that policies in the outer setting can shape the inner setting. The inner policies of most implementing organizations are determined by outer setting policies, which can affect the capacity for change, available budgets, and the freedom and involvement of individual stakeholders to be active in the implementation of interventions [69]. Important barriers related to the inner setting that have been identified previously [14] are, among others, competing priorities, lack of funding, lack of perceived responsibility among organizations, and limited capacity to take part in multiple initiatives. Those are mainly in line with the results of our study and are mostly related to the outer setting determinant of policies changing their focus from older adults to youth and from PA to other health goals like improving mental health and social interaction. To increase the implementation of PA interventions, intervention developers, therefore, need to make clear how PA interventions for older adults can contribute to those other health goals they aim to achieve. Implementation strategies like conducting local consensus discussions, conducting local needs assessments, and capturing and sharing knowledge are implementation strategies that are recommended to target these determinants [70,71].

A second relevant CFIR determinant is the *Relative priority* that the implementation of interventions receives. This is, besides changes in external policies, often influenced by changes in management. For example, high staff turnover and lack of support from leadership have also been identified previously as important implementation barriers within the inner setting [14]. Sometimes, the preferences of certain individuals may contradict the inner policies but may still affect the relative priority that an intervention gets, such as local politicians in municipalities that decide to implement interventions that best fit their personal interests or political agenda, which can affect the overall strategic direction. This may mean that support or prioritization of interventions is closely tied to the perspective of decision-makers. It, therefore, seems highly relevant when implementing interventions not only to talk to those who have an organizational role but also to those who have decision-making power [72,73]. 

The matter of *Compatibility* is similar to some of the issues described under intervention characteristics. Since every organization has different goals and is held accountable for these goals, the effects of an intervention are not always compatible with what is considered important in organizations. This also requires an inventory of what exactly a stakeholder who decides to implement an intervention wants to see reported as effects. Of course, this is also determined by the role of the stakeholder; a stakeholder who is only organizing may have different priorities regarding compatibility than those who are funding [74].

#### 4.2.4. Domain 4: Individual Characteristics 

CFIR states that organizations are made up of individuals and that the setting and intervention constructs all have a foundation in the behaviors of individuals. At this individual characteristics level, *Knowledge and beliefs about the intervention*, and *Other personal attributes* [31] were the main factors that influenced implementation. 

Regarding *Knowledge and beliefs about the intervention*, it was noteworthy that almost every stakeholder had doubts about what an intervention really is. Many did not seem to know the term, and when it was explained to them, they seemed to see interventions mostly as small-scale PA programs for groups. Part of this seemed to stem from a desire to use interventions that were already available in the region. However, these were often not proven effective interventions; instead, they were projects developed by local volunteers. It seems important, therefore, that discussions with stakeholders always start with a clear explanation of what intervention an intervention is, what different interventions there are, and why proven effective interventions are relevant.

Most stakeholders had a positive attitude to PA interventions and to older adults as the end user. However, not all stakeholders had decisional power, and as described earlier, a high staff turnover might also hinder intervention implementation as new staff members might have a less positive attitude. In line with the recommendations of Fernandez et al. (2019) [18], future initiatives are therefore recommended to involve multiple stakeholders per organization within their implementation plan: all stakeholders, including adopters, implementers, and those responsible for maintaining the intervention, should be identified.

### 4.3. Methodological Issues

Our study corroborates that CFIR provides a valuable framework for analyzing various aspects of intervention implementation while taking into account the variety and complexity of multiple organizations with different experiences. However, in our study, a key challenge was the lack of a specific intervention as an example. Without a concrete case study, it was sometimes difficult to directly translate CFIR’s theoretical framework accurately to our research questions. In addition, we found that because of the interconnectedness and complexity of the different domains within the CFIR framework, certain aspects in different CFIR domains sometimes seemed to overlap (e.g., the outer setting domain seemed to have a large impact on the inner setting domain and on the perceptions of the intervention characteristics). These methodological challenges suggest that while the CFIR Guide is highly useful, caution should be used when applying it to diverse contexts. It requires a careful and flexible approach to do justice to the unique situations and interconnectedness of different factors in each organization [75,76,77].

In this article, we have chosen not to demarcate determinants as either facilitating or hindering implementation, as the distinction between barriers and facilitators strongly differs from the point of view. For example, where the proven efficacy of interventions can be seen as a stimulating factor in terms of quality, it may also be seen as a hindering factor in terms of complexity. The perception of determinants is all related to a certain positive or negative context, which is all discussed in the current study. Labeling each determinant as a barrier or a facilitator would only complicate the findings of the current study. 

Another methodological issue is that in the current study, we only interviewed one (or sometimes two in one interview) stakeholder per organization. However, within an organization, the intervention adopter (i.e., the actual decision maker) might differ from the intervention implementer (i.e., the stakeholder who plays an active role in intervention implementation), and involving a broader perspective of stakeholders per organization is recommended. 

When interpreting the results of the present study, it should be kept in mind that we only interviewed stakeholders in the south of the Netherlands: as all countries may have different structures and different stakeholders involved, this should be taken into account when generalizing the results. However, this area in the Netherlands is in many ways comparable to other areas in Western Europe in terms of a similar population distribution and similar challenges in promoting PA in older adults. In addition, by including authors from three countries, we aimed to improve the international applicability of our findings. We would also like to mention that two of our researchers live and work in the area where this study was conducted, which may have introduced some interpretation bias. We tried to mitigate this potential bias by using a semi-structured interview guide based on theory, having our interview guide screened by an expert not tied to this area, having the coding and analyses performed by researchers from different regions, and adding two authors from other countries.

Our research team was all female, which may have introduced some gender bias into the study. However, apart from gender, the researchers were very different in other demographic characteristics such as age, experience, and country of residence, and therefore, we believe there was no need to specifically mitigate this limited chance of gender bias.

Another point worth mentioning is that qualitative research often identifies the type of organization that provided each quote. For privacy reasons, however, we were unable to include this information. For our findings, this omission is not problematic, as our analysis did not reveal substantial differences in opinions between different types of organizations. However, for future research, this may be a gap between science and practice, as in some cases, this information may be useful for creating an implementation plan tailored to different organizations. Replicating this study on a larger scale might solve privacy issues.

## 5. Conclusions

The current study identified relevant determinants among stakeholders when implementing PA interventions for older adults. Our research shows that the different CFIR domains seem heavily interrelated. Consequently, our conclusions and recommendations in one domain might influence the identified determinants in other domains as well. Overall, we recommend considering that:

A broad group of stakeholders is willing to play a role in implementation, but each has its own specific role: Ensure collaboration between stakeholders. Inner-setting determinants (e.g., goals and relative priority) and outer-setting determinants (mainly external policies) are highly interconnected: Stay attuned to national policies as these will affect the potential to implement PA interventions.In relation to internal and external policies, PA is often considered a means to an end. Regarding the intervention characteristics, make sure that when approaching stakeholders, the intervention is not just profiled as a PA intervention but as an intervention that can contribute to a broad perspective of health.The needs of the end user, especially of hard-to-reach populations, are a key outer setting determinant: Avoid one-size-fits-all approaches as an intervention characteristic.

The insight into implementation determinants, as provided in the current study, may provide relevant information that will contribute to increasing the potential public health impact of proven effective PA interventions. For future research, it is advisable to assess which implementation strategies can be applied to target these determinants. Although the evidence for implementation strategies is steadily developing (e.g., by the development of the Expert Recommendations of Implementation Change (ERIC) compilation [78]), these implementation strategies are sometimes seen as rather generic [79,80]. Using the Implementation Mapping (IM) protocol [18] can offer guidance, as it highlights the choice of appropriate behavior change techniques for influencing determinants, and it provides tools for tailoring implementation strategies focused on the determinants of individual stakeholders.

## Figures and Tables

**Table 1 geriatrics-09-00113-t001:** Summary of relevant determinants and key findings regarding implementation of physical activity (PA) interventions emerging from the interviews.

Determinants	Key Findings
**Domain 1: Intervention characteristics**	
Relative advantage	A preference exists for PA-interventions that do not have PA as the sole goal. but as a means to achieve broad health-related goals.
Evidence quality and strength	Proven effectiveness is not universally regarded as a critical factor for the acceptance of interventions.
	Evidence should not only relate to the intervention outcome itself (like increase in PA), but also to the engagement of participants, and particularly whether interventions successfully engage hard-to-reach populations.
	Stakeholders have diverse perspectives on the types of evidence that are necessary to evaluate interventions.
	There is a preference for qualitative measures over quantitative measures in evaluating outcomes.
	There is a desire to assess both short-term and long-term impacts of interventions.
Complexity and costs	There is a need for user-friendly manuals to facilitate the implementation of interventions.
	Interventions should be designed to minimize both labor and financial costs.
Adaptability and trialability	There is a recognized need to tailor interventions to the specific characteristics of the local context or to the end-user.
	It is considered beneficial to implement interventions on a small scale through pilot projects.
**Domain 2: Outer Setting**	
External policy and incentives	External policies (e.g., shifting focus to youth) are perceived to influence the inner setting decisions like relative priority.
Patient Needs	Stakeholders believe that alignment with the needs of end-users and the local context are important in implementation.
	Stakeholders believe that achieving sufficient PA is often not a primary goal for end-users, as they may often prioritize other concerns or have differing perceptions regarding their PA levels.
Cosmopolitism	No single stakeholder is capable of fulfilling all the roles required in the implementation process.
	Good communication and alignment within networks are regarded as important.
	Individual preferences of persons in other organizations are also perceived as relevant.
**Domain 3: Inner Setting**	
Implementation Climate	The inner setting implementation climate is often influenced by policies in the outer setting.
Relative Priority	Stakeholders mention that decision-makers in their organization are often other persons than those who implement an intervention, and perceived priority within organizations may change rather frequently over time.
Organizational incentives & rewards	Incentives and rewards vary between organizations, but the monitoring of goals is considered crucial throughout the intervention period.
Compatibility	The goals of the intervention need to be aligned with the goals of the stakeholder and the roles they can fulfil.
	Resources are limited and must be appropriately matched with the costs associated with the intervention.
**Domain 4: individual characteristics**	
Knowledge and beliefs	Stakeholders are unaware of what (proven effective) interventions, and their advantages and disadvantages are.
	There is a consensus that older adults are a relevant target population for interventions aimed at stimulating PA.
Other personal attributes	It is generally preferred that the target population should not be coerced into specific behaviors.
	There are differing opinions on whether and how digitalization should be incorporated into PA interventions for older adults.

## Data Availability

The data of this study has been made publicly available [82].

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
