# Peer review of "“Physical Activity Is Not the Answer to Everything, but It Is to a Lot”: Stakeholders’ Perceived Determinants of Implementing Physical Activity Interventions for Older Adults"

_geriatrics, 2024, doi:10.3390/geriatrics9050113_

Round 1

Reviewer 1 Report

Comments and Suggestions for Authors

This paper is the most interesting one I read in the last bit. It addresses a gap that scientits don't usually care about.

Can you address the following limits in page 15?

1- How are the two provinces in Netherleands representatives of others locations?

2- How having only female research team affected the results?

Author Response

Reviewer 1:

Comments and Suggestions for Authors

This paper is the most interesting one I read in the last bit. It addresses a gap that scientists don't usually care about.

We thank the reviewer for this very generous compliment, which we value greatly.

Can you address the following limits in page 15?

  • How are the two provinces in Netherlands representatives of others locations?

In general, these 2 provinces are very similar to many other Western European countries. For example, they have a mix of large cities and rural areas, and the challenges of implementing PA interventions for older adults in these two provinces are also largely comparable to the rest of the Netherlands: only one of the provinces has a slightly larger than average number of older adults compared to the rest of the Netherlands, but other population distributions such as socioeconomic status are quite similar to the average. As it is very difficult to find areas that are completely identical to the vast majority of other areas or countries, we tried to mitigate this potential bias by including authors from 3 different countries in Western Europe, thus including a perspective from other countries. The authors from countries other than the Netherlands indicated that there were no major differences between their countries and the locations included in the study.

We agree with the reviewer that we could improve how we explained this in the manuscript. We have now rewritten our text (in updated version of the manuscript now in lines 767-774): "When interpreting the results of the present study, it should be kept in mind that we only interviewed stakeholders in the south of the Netherlands: as all countries may have different structures and different stakeholders involved, this should be taken into account when generalizing the results. However, this area in the Netherlands is in many ways comparable to other areas in Western Europe regarding a similar population distribution and similar challenges in promoting PA in older adults. In addition, by including authors from three countries, we aimed to improve the international applicability of our findings.

  • How having only female research team affected the results?

Although our research team included only female researchers, we do not believe that this has introduced a significant bias in our study. We agree that the demographics of a research team can influence the study, but there are other demographics to consider besides gender. With this in mind, our research team has a wide range of experiences, perspectives, and expertise. For example, the ages of our team range from late twenties to early sixties, and the researchers have different focuses of expertise (e.g., DP focuses on implementing PA interventions for people over 50; JB focuses on PA and loneliness in the target population of people over 65 with mobility-impairing chronic conditions; GC focuses on health inequalities). In addition, the authors work in international collaborations, so they are also aware of factors such as cultural background, socioeconomic status, education, and personal experiences that may influence the results. In addition, we would like to mention that female researchers tend to be overrepresented in the discipline of health psychology research, so our research team is not so different from others.

We do agree with the author that we could pay more attention to this issue in our manuscript. Therefore, we have now rewritten the lines 781-784 to read: “Our research team was all female, which may have introduced some gender bias into the study. However, apart from gender, the researchers were very different in other demographic characteristics such as age, experience, and country of residence, and therefore we believe there was no need to specifically mitigate this limited chance of gender bias.”

Reviewer 2 Report

Comments and Suggestions for Authors

I have read and considered your paper carefully. The purpose of this study was to investigate the determinants of implementation of a physical activity intervention for community-dwelling older adults using a qualitative study design through stakeholder interviews. I believe that the qualitative study methodology was roughly appropriate. However, I have concerns about the control of bias affecting the results and interpretations.  The details are described below, please refer to this when revising the paper.    

Comments

1. Title

The title is very long. Please make it concise.

2.  Methods: Bias of the interviewers (the researchers)

The researcher works in the region where the study was conducted. I suspect that it appears to be an important bias in the analysis (coding) and may affect the results and interpretations. Because researcher may have preconception to the problems in the region and residents.  In addition, the research team was composed entirely of women. It is likely that gender differences in the perception and value of physical activity interventions cannot be excluded. This could also be an important bias in the analysis and interpretation. How were these biases controlled for in this study?

3.  Methods: Participant bias

In this study, participants were recruited using snowball sampling. However, perceptions and values of physical activity interventions are likely to vary widely by social position: policy maker, practitioner, health professional, social worker, etc. Similarly, they may differ by qualification, such as medical doctors, nurses, physiotherapists, social workers, etc. I believe that snowball sampling does not adequately reflect the diversity of values. In addition, from the above perspective, we believe that the analysis should take into account differences in qualifications and social positions.

4. Results: Intervention Characteristics

From the insight of the intervention characteristics described in the results section, I think that there is a discrepancy between the physical activity intervention for research purposes and that for implementation purposes. Is this due to a difference in values between researchers and practitioners? If so, I believe that researchers should be included stakeholders who was conducted the interviews.

5. Description of results and discussion

The amount of description in the Results and Discussion section is very large and may be difficult for readers to interpret. Please try to summarize the results of your thematic analysis and present them as concisely as possible.

6. References

Does the way you cite follow the "Instructions to Authors"?

Author Response

Reviewer 2:

Comments and Suggestions for Authors

 I have read and considered your paper carefully. The purpose of this study was to investigate the determinants of implementation of a physical activity intervention for community-dwelling older adults using a qualitative study design through stakeholder interviews. I believe that the qualitative study methodology was roughly appropriate. However, I have concerns about the control of bias affecting the results and interpretations.  The details are described below, please refer to this when revising the paper.

We thank the reviewer for the supportive comments and for the time invested to review our manuscript. We believe that by incorporating the feedback into our manuscript, it has gained in strength.    

  1. Title:

The title is very long. Please make it concise.

We suggest to change the title into: “Physical Activity is not the answer to everything, but it is to a lot”: stakeholders’ perceived determinants of implementing physical activity interventions for older adults.”

This way we still have a title with a quote that covers the content well, as is often done in qualitative research, and that includes all our key words.

  1. Methods: Bias of the interviewers (the researchers)

The researcher works in the region where the study was conducted. I suspect that it appears to be an important bias in the analysis (coding) and may affect the results and interpretations. Because researcher may have preconception to the problems in the region and residents. 

We understand the concerns of the reviewer that working in the region where the study was conducted, may have introduced bias into our findings. We would however like to highlight that not all researchers of this study work in the region where the study was conducted. KA and LS, who coded the interviews, are both not working in this province, as is RH who has reviewed the coding. In addition, we tried to mitigate this risk by having our interview guide being screened by a researcher who is not a part of our research team or authors, but is an expert in CFIR and works and lives in another area of the Netherlands (as mentioned in the method section and the acknowledgement): we feel this has resulted in an interview guide where no suggestions to the local situation were made. In addition, we included two authors who live and work in other countries.

This way, we believe we have taking precautions to mitigate this risk of bias, but agree with the reviewer that we cannot exclude that some bias may have entered our findings. Therefore we have made some alterations to the text:

First, we have now better explained the composition of our research team in in the Methods sector. Lines 112-117 now read: “The interviews were conducted in two provinces of the Netherlands (Limburg and Brabant): two of the researchers/authors involved in this study work in this region, the other researchers work in different regions of the Netherlands or abroad. The region were the interviews are conducted is largely comparable to the majority of Dutch provinces regarding demographic distributions, with only a slighter higher than average age distribution.” 

Second, we have added the next lines after line 775-780 to the Methodological issues (just before where we also now mention the potential risk of gender bias (see below)):  “We also like to mention that two of our  researchers live and work in the area where this study was conducted, which may have introduced some interpretation bias. We tried to mitigate this potential bias by using a semi-structured interview guide based on theory, by having our interview guide being screened by an expert not tied to this area, by having the coding and analyses being performed by researchers from different regions and by adding two authors from others countries.”

In addition, the research team was composed entirely of women. It is likely that gender differences in the perception and value of physical activity interventions cannot be excluded. This could also be an important bias in the analysis and interpretation. How were these biases controlled for in this study?

Although our research team included only female researchers, we do not believe that this would have affected our study. We agree that the demographics of a research team can influence the study, but there are other demographics to consider besides gender. With this in mind, our research team has a wide range of experiences, perspectives, and expertise. For example, the ages of our team range from late twenties to early sixties, and the researchers have different focuses of expertise (e.g., DP focuses on implementing PA interventions for people over 50; JB focuses on PA and loneliness in the target population of people over 65 with mobility-impairing chronic conditions; GC focuses on health inequalities). In addition, the authors work in international collaborations, so they are also aware of factors such as cultural background, socioeconomic status, education, and personal experiences that may influence the results. In addition, we would like to mention that female researchers tend to be overrepresented in the discipline of health psychology research, so our research team is not so different from others.

We agree with the author that we could pay more attention to this issue in our manuscript. Therefore, we have now added the lines 781-784: “Our research team was all female, which may have introduced some gender bias into the study. However, apart from gender, the researchers were very different in other demographic characteristics such as age, experience, and country of residence, and therefore we believe there was no need to specifically mitigate this limited chance of gender bias.”

  1. Methods: Participant bias

In this study, participants were recruited using snowball sampling. However, perceptions and values of physical activity interventions are likely to vary widely by social position: policy maker, practitioner, health professional, social worker, etc. Similarly, they may differ by qualification, such as medical doctors, nurses, physiotherapists, social workers, etc. I believe that snowball sampling does not adequately reflect the diversity of values. In addition, from the above perspective, we believe that the analysis should take into account differences in qualifications and social positions.

The reviewer raises an important point that values and perceptions might vary by social position, and that snowball sampling might not reflect this diversity. However,  would like to emphasize that our type of snowballing is different from what is commonly done in snowballing. We didn't use the typical type of snowballing where a participant is asked to suggest any other person that may want to participate in the study. What we did was to ask each participant to name another type of stakeholder that they thought should be involved. We explained that by another type of stakeholder, we meant other organizations that might have other qualifications, other objectives for interventions, other roles in society, etc., i.e. we specifically asked them to name any type of organization that they thought had a role or should have a role in implementation of PA interventions for older adults. In this way, we aimed to include all types of stakeholders that other stakeholders considered important in implementation. The stakeholders we interviewed come from a wide range of organizations, as described on pages 207-221, reflecting different social positions, different qualifications, different gender, age, experience, etc. We found that the opinions of the interviewees were generally quite consistent: it was therefore not possible or necessary in the analyses to take into account differences in, for example, social position or qualifications of stakeholders.

Upon reading the reviewer comment, we have checked if literature use other terms for this specific type of snowballing, and we now found that this type of snowballing is more commonly known as “referral sampling” ( reference: Rapley, T. (2013). Sampling strategies in qualitative research. In U. Flick (Ed.), The SAGE Handbook of Qualitative Data Analysis (pp. 49-63). SAGE Publications Ltd. https://dx.doi.org/10.4135/9781446282243). We therefore now have used this term in the text and also explain better what we have done.

First in line 109-111 we have modified the text into: “Stakeholders were recruited by referral sampling: in referral sampling participants are asked to recommend organizations that are particularly relevant focusing on those with specific expertise or involvement in the area of interest.”

And lines 126-133: “After each interview, stakeholders were asked to identify other stakeholders that they considered important for the implementation of PA interventions for older adults in the Netherlands. We didn't use a traditional snowball method, where interviewees are asked to suggest another person, but we used referral sampling by explicitly asking what kind of other organizations or people with other kinds of roles are important in the implementation of PA interventions. In this way, we aimed to include all potentially relevant stakeholders within a wide range of positions or qualifications.”

  1. Results: Intervention Characteristics

From the insight of the intervention characteristics described in the results section, I think that there is a discrepancy between the physical activity intervention for research purposes and that for implementation purposes. Is this due to a difference in values between researchers and practitioners? If so, I believe that researchers should be included stakeholders who was conducted the interviews.

We are not sure exactly what the reviewers means with this comment; especially the last sentence seems to leave some room for interpretation. We agree that researchers may have other values and interests when designing or evaluating interventions in RCT’s, than stakeholders who want to implement an intervention in a real world setting (i.e. after the RCT). We addressed this issue on several points in the manuscript, but upon reading the reviewer’s comment we tried to enhance this in the manuscript.

In lines 535-543 we have now emphasized that many PA interventions show a primary focus on increasing PA, which from a researchers perspective may be interesting, but that we found that stakeholders have another perspective and feel that PA is very relevant, but not only in itself but to reach other goals. These lines now read:

“Although researchers are well aware of the comprehensive health benefits of physical activity (PA), it's important to recognize that stakeholders may not view PA as a sufficient standalone goal. They understand that PA promotes health but may narrowly associate it with just increasing activity levels. Therefore, when designing and implementing a PA intervention, it is crucial to communicate clearly that PA is a means to achieve a broader spectrum of health benefits, including physical, mental, and social well-being. This approach ensures that the intervention is not merely perceived as promoting more physical activity but is valued as a holistic strategy for enhancing overall health.”

In addition, we also emphasized this now in lines 566-574 which now read:

“Regarding the CFIR determinant Evidence Quality & Strength, our study highlights a discrepancy between how efficacy of an intervention is determined in randomized controlled trials (RCTs) and how stakeholders define the success of an intervention. While intervention developers typically prioritize the effectiveness of significantly increasing physical activity levels, stakeholders may have different priorities. Although they appreciate the quality assurance that proven effectiveness offers, stakeholders might be more interested in other outcomes. For example, some stakeholders are primarily concerned with reaching a large number of end-users, whereas others find it successful when hard-to-reach populations are reached, such as populations of lower socio-economic positions.”

Other changes we made are in 3.2.4 Knowledge & Beliefs, lines 445-453 which now read:

“During the interviews, when we asked stakeholders about the role they could play in implementing interventions, it became apparent that many were unfamiliar with the term 'intervention' and often requested an explanation or definition. Initially, most stakeholders associated interventions with small-scale, locally developed programs conducted in group settings. However, after providing a definition (“proven effective scientifically developed programs aiming at changing individual PA behavior) and an example (as mentioned in the Methods), stakeholders understood wat was meant. They generally expressed a positive attitude towards the health benefits of promoting physical activity (PA) and the target population of older adults.”

Additionally, we rephrased lines 188-195 to now read: "When discussing the domain of intervention characteristics, we used a proven effective PA online computer-tailored intervention as an example [22-24]. This intervention was chosen as a good representation of a scientifically developed and effective intervention capable of large-scale implementation. It also illustrates a type of intervention that may not immediately come to mind for all interviewed stakeholders, who may have previously focused on small-scale local programs."

  1. Description of results and discussion

The amount of description in the Results and Discussion section is very large and may be difficult for readers to interpret. Please try to summarize the results of your thematic analysis and present them as concisely as possible.

We agree with the reviewer that some sort of summarization could enhance readability. Of course, qualitative research often requires elaborate descriptions to adequately capture the rich insights provided by the study, and our article is in length comparable to most qualitative studies. To address the reviewers suggestion for some sort of summarization, we propose keeping the main body of the text unchanged to preserve the detailed insights, conclusions, and recommendations. We propose to add a table to the Results, summarizing the main findings, which can then be referenced in the Discussion. This table would serve as a quick reference guide, improving readability without compromising the depth of all relevant content. This approach also aligns with the other reviewer's preference for only minor changes, ensuring that we maintain the integrity and comprehensive nature of the qualitative analysis. The table will allow readers to easily cross-reference the results with the discussion, facilitating a smoother interpretation process. We have now added this table at the end of the Results section. In the Discussion section we now mention that for each determinant that emerged we give an overview of the main findings. We feel that in this way we have found a good balance between giving all relevant content while in the same time enhancing readability.

  1. References

Does the way you cite follow the "Instructions to Authors"?

Yes, we adhere both to the journals instructions as to the APA guidelines on reporting qualitative information.

Comments Reviewer 1:

This paper is the most interesting one I read in the last bit. It addresses a gap that scientists don't usually care about.

We thank the reviewer for this very generous compliment, which we value greatly.

Can you address the following limits in page 15?

How are the two provinces in Netherlands representatives of others locations?

In general, these 2 provinces are very similar to many other Western European countries. For example, they have a mix of large cities and rural areas, and the challenges of implementing PA interventions for older adults in these two provinces are also largely comparable to the rest of the Netherlands: only one of the provinces has a slightly larger than average number of older adults compared to the rest of the Netherlands, but other population distributions such as socioeconomic status are quite similar to the average. As it is very difficult to find areas that are completely identical to the vast majority of other areas or countries, we tried to mitigate this potential bias by including authors from 3 different countries in Western Europe, thus including a perspective from other countries. The authors from countries other than the Netherlands indicated that there were no major differences between their countries and the locations included in the study.

We agree with the reviewer that we could improve how we explained this in the manuscript. We have now rewritten our text (in updated version of the manuscript now in lines 767-774): "When interpreting the results of the present study, it should be kept in mind that we only interviewed stakeholders in the south of the Netherlands: as all countries may have different structures and different stakeholders involved, this should be taken into account when generalizing the results. However, this area in the Netherlands is in many ways comparable to other areas in Western Europe regarding a similar population distribution and similar challenges in promoting PA in older adults. In addition, by including authors from three countries, we aimed to improve the international applicability of our findings.

How having only female research team affected the results?

Although our research team included only female researchers, we do not believe that this has introduced a significant bias in our study. We agree that the demographics of a research team can influence the study, but there are other demographics to consider besides gender. With this in mind, our research team has a wide range of experiences, perspectives, and expertise. For example, the ages of our team range from late twenties to early sixties, and the researchers have different focuses of expertise (e.g., DP focuses on implementing PA interventions for people over 50; JB focuses on PA and loneliness in the target population of people over 65 with mobility-impairing chronic conditions; GC focuses on health inequalities). In addition, the authors work in international collaborations, so they are also aware of factors such as cultural background, socioeconomic status, education, and personal experiences that may influence the results. In addition, we would like to mention that female researchers tend to be overrepresented in the discipline of health psychology research, so our research team is not so different from others.

We do agree with the author that we could pay more attention to this issue in our manuscript. Therefore, we have now rewritten the lines 781-784 to read: “Our research team was all female, which may have introduced some gender bias into the study. However, apart from gender, the researchers were very different in other demographic characteristics such as age, experience, and country of residence, and therefore we believe there was no need to specifically mitigate this limited chance of gender bias.”

Round 2

Reviewer 2 Report

Comments and Suggestions for Authors

I believe that the revised manuscript is appropriately addressed and modified to the reviewers' comments. The results of the study may be interesting topic that will provide useful information to practitioners and researchers.

Author Response

We have included the suggested articles in paragraph one (highlighted in yellow).